METHODS ARTICLE

# Identifying patterns differing between high-dimensional datasets with generalized contrastive PCA

**Eliezyer Fermino de Oliveira**[1], **Pranjal Garg**[2], **Jens Hjerling-Leffler**[3], **Renata Batista-Brito**[1,4,5], **Lucas Sjulson**[1,4]*

**1** Dominick P. Purpura Department of Neuroscience, Albert Einstein College of Medicine, Bronx, New York, United States of America, **2** All India Institute of Medical Sciences, Rishikesh, India, **3** Department of Medical Biochemistry and Biophysics, Karolinska Institutet, Stockholm, Sweden, **4** Department of Psychiatry and Behavioral Sciences, Albert Einstein College of Medicine, Bronx, New York, United States of America, **5** Department of Genetics, Albert Einstein College of Medicine, Bronx, New York, United States of America

\* luke@sjulsonlab.org

**Data availability statement:** The data used in this paper were taken from different sources. The facial expression dataset is a subset of the

## Abstract

High-dimensional data have become ubiquitous in the biological sciences, and it is often desirable to compare two datasets collected under different experimental conditions to extract low-dimensional patterns enriched in one condition. However, traditional dimensionality reduction techniques cannot accomplish this because they operate on only one dataset. Contrastive principal component analysis (cPCA) has been proposed to address this problem, but it has seen little adoption because it requires tuning a hyperparameter resulting in multiple solutions, with no way of knowing which is correct. Moreover, cPCA uses foreground and background conditions that are treated differently, making it ill-suited to compare two experimental conditions symmetrically. Here we describe the development of generalized contrastive PCA (gcPCA), a flexible hyperparameter-free approach that solves these problems. We first provide analyses explaining why cPCA requires a hyperparameter and how gcPCA avoids this requirement. We then describe an open-source gcPCA toolbox containing Python and MATLAB implementations of several variants of gcPCA tailored for different scenarios. Finally, we demonstrate the utility of gcPCA in analyzing diverse high-dimensional biological data, revealing unsupervised detection of hippocampal replay in neurophysiological recordings and heterogeneity of type II diabetes in single-cell RNA sequencing data. As a fast, robust, and easy-to-use comparison method, gcPCA provides a valuable resource facilitating the analysis of diverse high-dimensional datasets to gain new insights into complex biological phenomena.

Chicago Face Dataset. We are not allowed to re-distribute this dataset, but it is publicly available at https://www.chicagofaces. org/download/, publication DOI: https://doi.org/10.3758/s13428-014-0532-5. The hippocampal dataset is publicly available at buzsakilab.com/wp/database/, (DOI: https://doi.org/10.5281/zenodo.4307883). We provide a file with the minimal dataset to replicate our analysis alongside the analysis code (DOI: 10.5281/zenodo.13308211, link: https://zenodo.org/doi/10.5281/zenodo. 13308211); this file contains identifiers of the animal number and session. The pancreatic single-cell RNA sequencing dataset is publicly available at the Gene Expression Omnibus (GEO), with accession code GSE153855, link: https://www.ncbi.nlm.nih.gov/geo/query/acc. cgi?acc=GSE153855. Other files to replicate our analysis are provided alongside the analysis code. The code to replicate the analysis in this manuscript is available on GitHub (link: https: //github.com/eliezyer/deOliveira_gcPCA_2024 DOI: https://zenodo.org/doi/10.5281/ zenodo.13308211). Code to replicate the preprocessing of the Chicago Face Dataset and the hippocampal electrophysiology dataset is also provided. The gcPCA toolbox is publicly available at https://github.com/SjulsonLab/ generalized_contrastive_PCA.

**Funding:** This work was supported by funds from the National Institute on Drug Abuse (https://nida.nih.gov/, DP1 DA051608 and R01 DA051652), as well as from the Whitehall (http://www.whitehall.org/), Keck (http://www.wmkeck.org), and McManus Foundations, to LS. The funders had no role in study design, data collection and analysis, decision to publish, or preparation of the manuscript.

**Competing interests:** The authors have declared that no competing interests exist.

## Author summary

Technological advances in the biological sciences have led to the proliferation of large, complex datasets for which analysis is challenging. Analyses for these datasets rely heavily on dimensionality reduction techniques, which extract reduced-complexity representations of the data that are easier to analyze and interpret. However, these techniques typically operate on only one dataset, and many biological experiments involve comparing two datasets collected under different conditions. Contrastive principal components analysis (cPCA) was previously developed for this purpose, but it has limitations that have precluded its widespread adoption. Here we introduce generalized contrastive principal components analysis (gcPCA), a method that overcomes these limitations. We first explain the mathematical basis of gcPCA, then describe an open-source gcPCA toolbox with implementations in Python and MATLAB. Finally, we demonstrate the utility of gcPCA in analyzing diverse biological datasets, highlighting its versatility as a tool to compare experimental data collected under two different conditions.

## Introduction

Investigators in the biological sciences are increasingly collecting high-dimensional datasets that are challenging to analyze, with modalities ranging from imaging to electrophysiology to single-cell RNA sequencing. Dimensionality reduction algorithms such as principal components analysis (PCA) and its many variants [1–5] are used widely to help simplify these datasets and facilitate analysis. PCA examines the covariance structure of the data to find dimensions that account for more variance than chance; these constitute patterns that are overrepresented in the data, such as assemblies of neurons whose activity fluctuates up and down together across time in a neural recording [6–9], or networks of genes that are up- or down-regulated together across cells in a single-cell RNAseq dataset [10,11]. However, in many cases, the goal is to compare data collected under two different experimental conditions, which we refer to here as datasets. Since PCA and other dimensionality reduction techniques operate on only one dataset, they cannot take experimental conditions into account.

The most common approach for comparing two high-dimensional datasets is linear discriminant analysis (LDA) [12] or its multidimensional analog, partial least squares discriminant analysis [13]. These methods find dimensions that optimally distinguish one dataset from the other, which could correspond to which neurons fire more, or which genes are upregulated, in condition $A$ vs. condition $B$. However, an analogous method to compare the covariance structure of two datasets is not as well established. This addresses more subtle and detailed questions, such as which subsets of neurons exhibit increased temporal correlations in condition $A$ than $B$, or which subsets of genes are more likely to be up- or downregulated together in individual cells in condition $A$ than $B$. Mathematically, answering these questions corresponds to finding dimensions that account for more variance in $A$ than in $B$ (S1A Fig).

Recently, contrastive PCA (cPCA) was proposed as a method to address this problem [14]. Although cPCA is an important first step, it requires a hyperparameter $\alpha$, which controls how much covariance from the second condition to subtract from the first. The algorithm must therefore iterate over multiple choices of $\alpha$ with no objective criteria to determine which value of $\alpha$ yields the correct answer. Moreover, cPCA is asymmetric, identifying the most enriched dimensions in the first condition after subtracting out the second condition as background; it cannot treat the two experimental conditions equally.

Here we propose a novel solution to these problems we call generalized contrastive PCA (gcPCA). We first demonstrate the role the $\alpha$ hyperparameter plays in cPCA, then explain our strategy for eliminating it. We then describe an open source toolbox for Python and MATLAB implementing several versions of gcPCA with different objective functions that are either asymmetric or symmetric, orthogonal or non-orthogonal, or sparse or dense, tailored to suit the specific application at hand. Finally, we demonstrate the utility of gcPCA in the analysis of diverse biological datasets.

## Results

### The cPCA hyperparameter $\alpha$ compensates for bias toward high-variance dimensions in noisy, finitely-sampled datasets

To explain the need for the hyperparameter $\alpha$ in cPCA and how we avoid it in gcPCA, we will describe the objective function of each method and show how they perform in generated synthetic data. For illustration purposes, we generated synthetic data for two experimental conditions containing two-dimensional manifolds on a background of high-variance shared dimensions. The generated data consisted of condition *A*, with a manifold (additional variance) in the 71st and 72nd dimensions (ranked in order of descending variance), and condition *B*, with a manifold in the 81st and 82nd dimensions (Fig 1A). The manifold dimensions contained less total variance than most of the other dimensions in the dataset, but their variance is two-fold higher in one condition relative to the other (i.e., the 81st and 82nd dimensions have twice as much variance in condition *B* than condition *A*).

A key property of real-world biological datasets is that they are noisy and finitely sampled. We aimed to model the finite data regime by comparing $1 \times 10^3$ samples (finite data) to $1 \times 10^5$ samples, which approximates infinite data. To inspect the effects of finite sampling on estimated variance, we projected the "finite" and "infinite" data onto the ground truth dimensions and calculated the variance explained by each (see methods). Our results (Fig 1B) reveal that finite sampling yields noisy estimates of the true variance, with greater noise in high-variance dimensions.

To understand the practical consequences of this, it is helpful to start by reviewing traditional PCA. With PCA, the principal components of a data matrix $\mathbf{D}$, of size $n \times p$ (samples $\times$ features), are the dimensions explaining the most variance. These can be identified by estimating the covariance matrix $\mathbf{C} = \mathbf{D}^\mathsf{T}\mathbf{D}/(n-1)$, then solving the following quadratic optimization problem in equation 1:

$$\underset{\mathbf{x}\,:\,\mathbf{x}^\mathsf{T}\mathbf{x}=1}{\arg\max} \quad \mathbf{x}^\mathsf{T}\mathbf{C}\mathbf{x} \tag{1}$$

This problem can be solved by eigendecomposition of $\mathbf{C}$, yielding the matrix of eigenvectors $\mathbf{X}$ known as principal components (PCs).

To extend this to two datasets, a logical strategy involves formulating an objective function to describe the difference in variances between the two conditions, enabling us to extract dimensions that show the greatest increase in variance in *A* relative to *B*. We now have two covariance matrices, $\mathbf{C}_A$ and $\mathbf{C}_B$, and the contrastive PCs (cPCs) are the vectors that maximize the objective function 2:

$$\underset{\mathbf{x}\,:\,\mathbf{x}^\mathsf{T}\mathbf{x}=1}{\arg\max} \quad \mathbf{x}^\mathsf{T}(\mathbf{C}_A - \mathbf{C}_B)\mathbf{x} \tag{2}$$

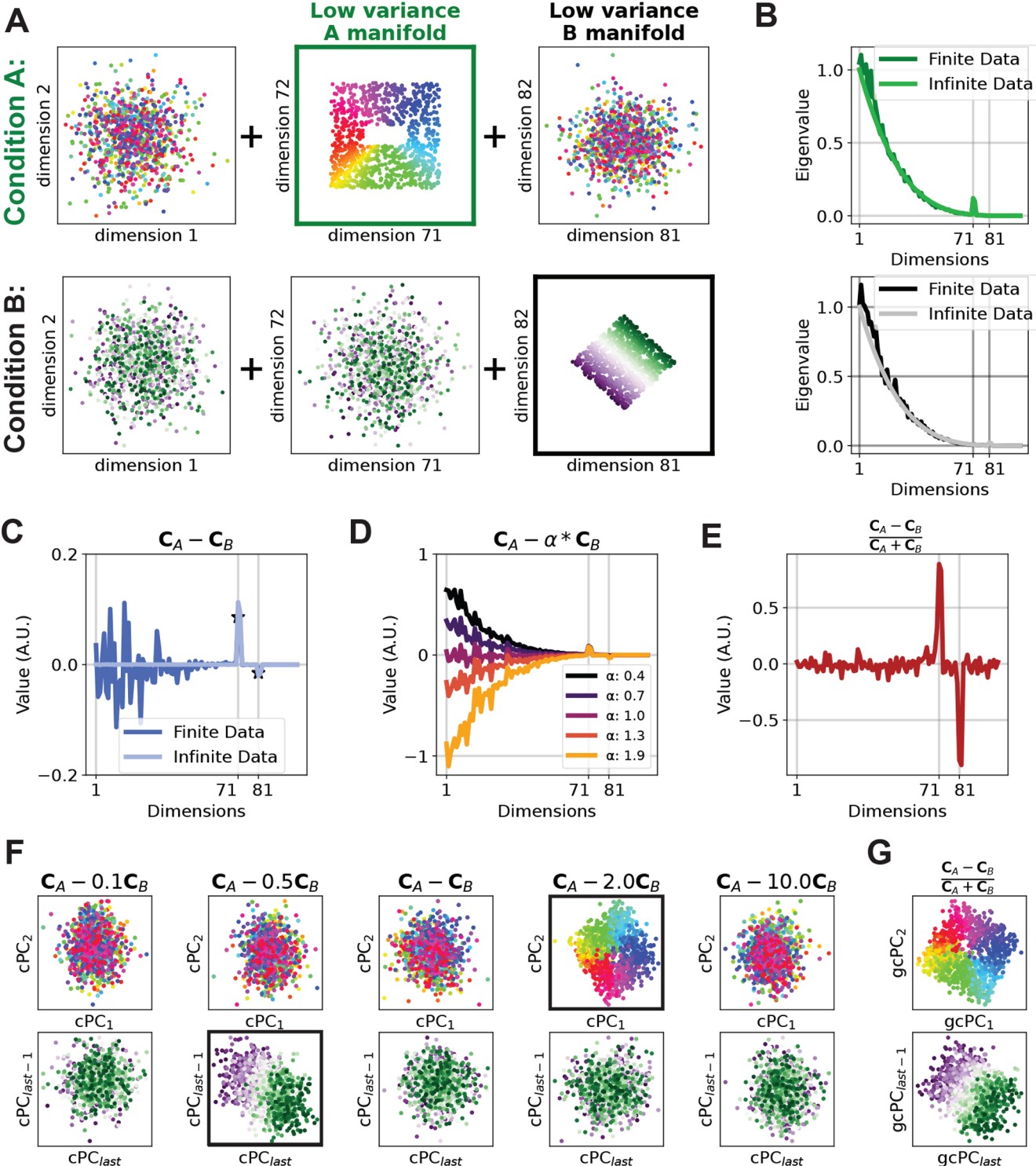

**Fig 1. Generalized contrastive PCA avoids cPCA's need for the hyperparameter $\alpha$ in noisy, finitely-sampled data. (A)** We generated two conditions in noisy synthetic data that each contain low-variance manifolds that are not present in the other. These manifolds have lower overall variance than many other dimensions and are not trivially discoverable. **(B)** Eigenvalue spectra for each condition estimated from finite (dark line) or infinite (light line) data. Note the sampling error in the finite date case. **(C)** With infinite data, eigendecomposition of $(\mathbf{C}_A - \mathbf{C}_B)$ suffices to extract the correct answers (dimensions 71-72 and 81-82, light lines). However, with finite data, these peaks are smaller than the sampling error in high-variance dimensions, creating a bias toward high-variance dimensions being selected. **D** cPCA uses the hyperparameter $\alpha$ to adjust how much influence $\mathbf{C}_B$ has on $\mathbf{C}_A$. As $\alpha$ increases, the bias toward high-variance dimensions decreases until it becomes negative with $\alpha > 1$, eventually exposing the differences in lower-variance dimensions. Importantly,

there is no way to know which value of $\alpha$ yields the correct solution. **(E)** Using gcPCA, the dimensions most changed in each condition are identified correctly, even with finitely-sampled data. **(F)** cPCA with the optimal choice of $\alpha$ does not extract the correct dimensions in *B*. **(G)** gcPCA identifies the enriched dimensions in each condition and correctly returns the low-variance manifolds. Because gcPCA is symmetric, it extracts the correct dimensions in both *A* and *B*.

Analogous to traditional PCA, this problem can be solved by eigendecomposition of $(\mathbf{C}_A - \mathbf{C}_B)$. This yields cPCs that account for more variance in either condition *A* or *B*, corresponding to the eigenvectors with the largest or smallest eigenvalues, respectively. With infinite data, the precise estimates of the eigenspectrum correctly find the dimensions enriched in conditions *A* and *B* (Fig 1C, light line), but with finite data, it fails to do so because the sampling error in higher-variance dimensions is larger than the true signal in the lower-variance dimensions (Fig 1C, dark line). In other words, it has a systematic bias toward high-variance dimensions. To compensate for this effect, cPCA [14] introduces the hyperparameter $\alpha$, changing the following objective function to 3:

$$\underset{\mathbf{x}\,:\,\mathbf{x}^\mathsf{T}\mathbf{x}=1}{\arg\max} \quad \mathbf{x}^\mathsf{T}(\mathbf{C}_A - \alpha\mathbf{C}_B)\mathbf{x} \tag{3}$$

As discussed in [14], $\alpha$ represents a trade-off determining the extent to which $\mathbf{C}_B$ influences the identification of enriched vectors in $\mathbf{C}_A$. In our synthetic data, we can visually appreciate the effect of different values of $\alpha$ in the resulting cPCA objective function value (Fig 1D). In effect, $\alpha$ tunes the amount of bias toward high-variance dimensions in the cPCA calculation, with $\alpha < 1$ biasing toward high-variance dimensions and $\alpha > 1$ biasing against them. If the correct value of $\alpha$ is selected, the sampling error in the high-variance dimensions of *A* and *B* will cancel out, yielding the correct answer. In this example, $\alpha = 2$ yields the correct solution that dimensions 71-72 are enriched in *A* (Fig 1F), but other values of $\alpha$ yield equally plausible, but incorrect, solutions. Importantly, we can only determine which solution is correct because we knew the answer in advance, which is not typically the case for experimental data. Further, negative values in cPCA are generally interpreted as dimensions enriched in condition *B* [14,15], but our simulation shows that values of $\alpha$ larger than 1 bias the highest-variance dimensions to be negative (Fig 1D). This creates the illusion that these dimensions are enriched in condition *B*, even though the correct answer is that only dimensions 81-82 are enriched in *B*. Similar to the situation of finding dimensions enriched in *A*, the results depend on the choice of $\alpha$, with no way to determine which solution is correct (Fig 1F). The range of $\alpha$'s yielding the correct solution can also be incredibly narrow, as in S2 Fig, where $\alpha = 2.6$ yields the correct solution, but 2.2 or 3.0 do not.

## gcPCA avoids hyperparameters by including a normalization factor to penalize high-variance dimensions

Our goal for gcPCA was to eliminate the need for hyperparameters and provide unique, correct solutions. To mitigate the bias toward high-variance dimensions, we therefore introduced a normalization factor to penalize high-variance dimensions. gcPCA can use several different normalization factors depending on the task at hand (Table 1); here we will use the total variance in both conditions, which can be calculated by summing the covariance matrices $(\mathbf{C}_A + \mathbf{C}_B)$. The objective function then becomes (eq. 4):

$$\underset{\mathbf{x}\,:\,\mathbf{x}^\mathsf{T}\mathbf{x}=1}{\arg\max} \quad \frac{\mathbf{x}^\mathsf{T}(\mathbf{C}_A - \mathbf{C}_B)\mathbf{x}}{\mathbf{x}^\mathsf{T}(\mathbf{C}_A + \mathbf{C}_B)\mathbf{x}} \tag{4}$$

**Table 1. gcPCA variants in the gcPCA toolbox.**

| gcPCA version | Symmetric | Orthogonal | Sparse solution | Note |
|---|---|---|---|---|
| v1 | ✗ | ✓ | ✓ | Equivalent to cPCA |
| v2 | ✗ | ✗ | ✓ | Objective function 5 |
| v2.1 | ✗ | ✓ | ✗ | Objective function 5 |
| v3 | ✗ | ✗ | ✓ | Objective function 6 |
| v3.1 | ✗ | ✓ | ✗ | Objective function 6 |
| v4 | ✓ | ✗ | ✓ | Objective function 7 |
| v4.1 | ✓ | ✓ | ✗ | Objective function 7 |

This problem can be solved by eigendecomposition of $\mathbf{M}^{-1}(\mathbf{C}_A - \mathbf{C}_B)\mathbf{M}^{-1}$, where $\mathbf{M}$ is the matrix square root of the denominator $(\mathbf{C}_A + \mathbf{C}_B)$ (see Methods). The resulting generalized contrastive PCs (gcPCs) maximize relative, rather than absolute, changes in variance between conditions $A$ and $B$. Because the penalty and the sampling error both scale with the variance, this effectively handles the bias toward high-variance dimensions and successfully extracts the ground truth dimensions in our synthetic data, even with finite sampling (Fig 1E, 1G), and even when the range of acceptable $\alpha$ is narrow (S2 Fig).

This creates two minor complications to be aware of: first, unlike PCs or cPCs, gcPCs are not orthogonal in the original feature space. Instead, they are orthogonal in the normalized feature space in which high-variance dimensions have already been penalized by multiplication with $\mathbf{M}^{-1}$. For the purpose of contrasting $A$ and $B$, this is usually the best approach. However, if orthogonality in the original feature space is important for a particular application, we have implemented versions of gcPCA with an orthogonality constraint (S3A-S3D Fig, see Methods). Second, the normalization factor can create numerical instability if the data contain dimensions with zero or near-zero variance in the denominator, in effect leading to division by zero. This occurs if there are fewer samples than features or if the features are strongly correlated. Our implementation of gcPCA solves this problem by detecting and excluding zero-variance dimensions before performing the calculation (see Methods).

### The open-source gcPCA toolbox contains multiple gcPCA variants enabling optimal handling of diverse use cases

We have developed an open-source gcPCA toolbox with implementations in Python and MATLAB of several different variants of gcPCA. This toolbox is freely available at: https://github.com/SjulsonLab/generalized_contrastive_PCA. Here we will present the different variants of gcPCA and their use cases.

**gcPCA v1: traditional cPCA.** For version 1, we include an implementation of the original cPCA algorithm that finds cPCs maximizing the objective function in Eq. 3.

**gcPCA v2: gcPCA maximizing A/B.** Here we include an implementation that finds gcPCs maximizing the ratio of variance in A to B:

$$\underset{\mathbf{x}\,:\,\mathbf{x}^\mathsf{T}\mathbf{x}=1}{\arg\max} \quad \frac{\mathbf{x}^\mathsf{T}\mathbf{C}_A\mathbf{x}}{\mathbf{x}^\mathsf{T}\mathbf{C}_B\mathbf{x}} \tag{5}$$

Like cPCA, this method is asymmetrical, meaning it is suitable for situations in which $A$ is a foreground condition and $B$ is a background condition; in other words, $A$ is presumed to be equal to $B$ plus a low-dimensional pattern, and the goal is to extract that pattern. The

resulting eigenvalues are the ratio of the variance a given gcPC accounts for in *A* to the variance it accounts for in *B*. Thus, they fall in the range $[0, \infty)$, with gcPCs enriched in *A* having eigenvalues > 1.

**gcPCA v3: gcPCA maximizing (A-B)/B.** The second method developed is also asymmetrical but based on a relative change:

$$\underset{\mathbf{x}:\mathbf{x}^\mathsf{T}\mathbf{x}=1}{\arg\max} \quad \frac{\mathbf{x}^\mathsf{T}(\mathbf{C}_A - \mathbf{C}_B)\mathbf{x}}{\mathbf{x}^\mathsf{T}(\mathbf{C}_B)\mathbf{x}} \tag{6}$$

This method is closely-related to v2 and is suitable for scenarios in which the investigator wishes to define the gcPCs based on a relative change to a background condition (i.e., finding a 30% increase in the variability of neural activity in condition *A* relative to *B*). The eigenvalues returned are in the range $[-1, \infty)$, with gcPCs enriched in *A* having eigenvalues > 0.

**gcPCA v4: gcPCA maximizing (A-B)/(A+B).** The last of the three methods is based on a relative change:

$$\underset{\mathbf{x}:\mathbf{x}^\mathsf{T}\mathbf{x}=1}{\arg\max} \quad \frac{\mathbf{x}^\mathsf{T}(\mathbf{C}_A - \mathbf{C}_B)\mathbf{x}}{\mathbf{x}^\mathsf{T}(\mathbf{C}_A + \mathbf{C}_B)\mathbf{x}} \tag{7}$$

This method is symmetrical, treating conditions *A* and *B* equally, and is appropriate for contrasting conditions in which *B* is a distinct condition and not merely a background to be removed, for example comparing neural data in sleep vs. wakefulness. The eigenvalues are in the range $[-1, 1]$ and are easily interpretable as a traditional index of the form $(A - B)/(A + B)$: 1 means that a gcPC only accounts for variance in *A*, -1 means it only accounts for variance in *B*, and 0 means it accounts for equal variance in both. This method is fully symmetrical in the sense that switching *A* and *B* will yield the same gcPCs with the signs of the eigenvalues reversed.

**gcPCA v2.1, v3.1, and v4.1: Orthogonal gcPCA.** Unlike PCs or cPCs, gcPCs are not orthogonal in the original feature space, but rather in the feature space after normalization has been applied to penalize high-variance dimensions (S3A and S3C Fig). Because orthogonality in the original feature space may be important for some applications, we also include versions of gcPCA with an orthogonality constraint (see Methods). These provide orthogonal solutions at the cost of an increase in processing time (S3A–S3D Fig). gcPCA v2.1 is the orthogonal version of v2, v3.1 is the orthogonal version of v3, and v4.1 is the orthogonal version of v4.

**Sparse vs. dense gcPCA.** In high-dimensional datasets, it is often desirable to perform feature selection for easy interpretation of the results. With that in mind, we have also developed sparse versions of gcPCA based on sparse PCA [3,16]. This method works by first finding the gcPCs based on the objective function selected, then performing feature selection using an $L1$ lasso penalty (see Methods). Unfortunately, this cannot be reduced to an eigenvalue problem and must rely on gradient descent, increasing the processing time. Optimal sparse solutions generally are not orthogonal (S3E and S3F fig), and they cannot be used with the orthogonality constraint.

## Unsupervised extraction of facial expression features with gcPCA

To illustrate the utility of gcPCA with real-world datasets, we first use the Chicago Face Dataset [17], which contains faces with different facial expressions. Here we used happy and angry expressions as condition *A* and neutral faces as condition *B* (Fig 2A). This dataset is

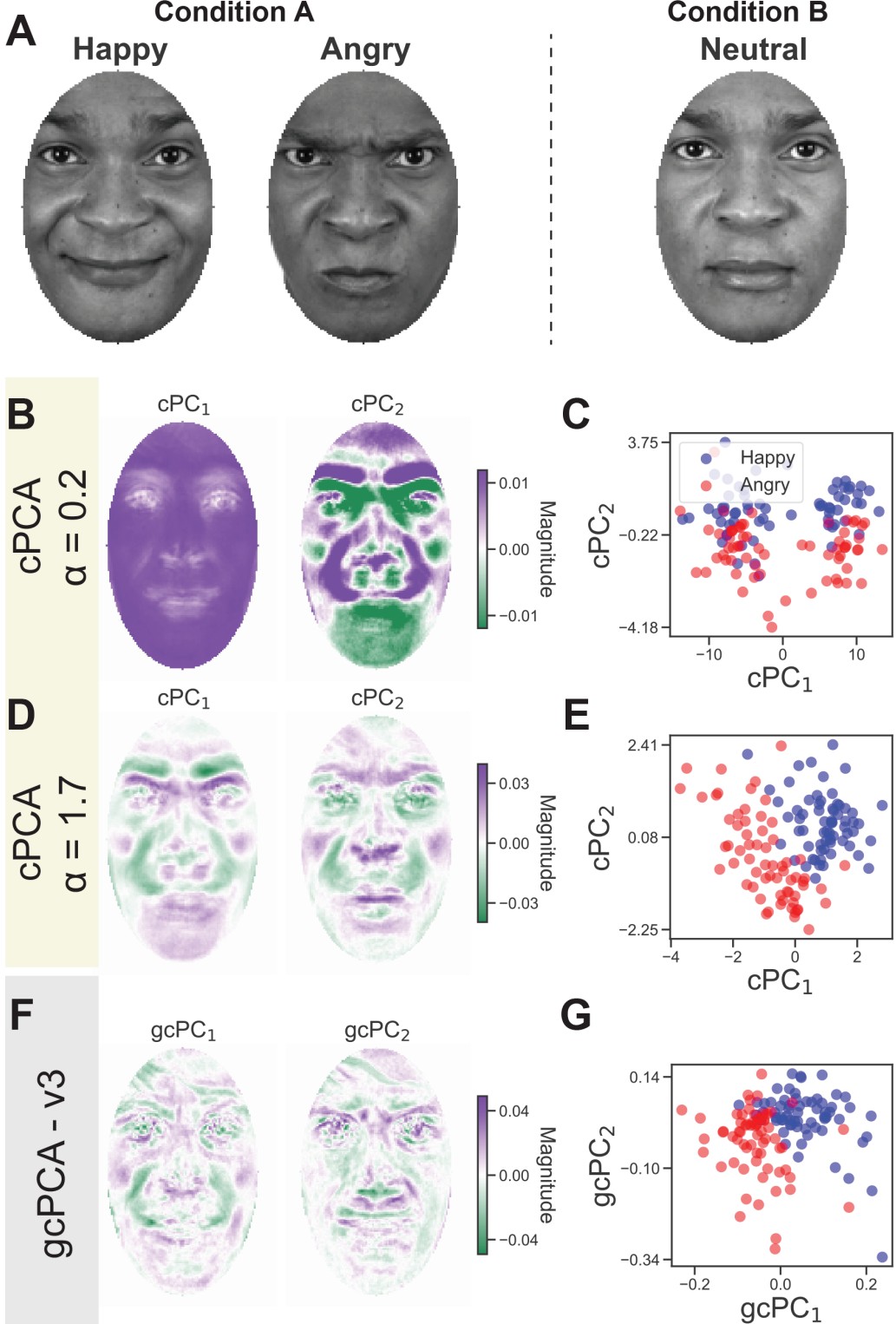

**Fig 2. gcPCA correctly extracts contrastive facial features in the Chicago Face Dataset. (A)** For this test, contrastive methods were applied to a set of happy and angry face images (Condition *A*) versus neutral face images from the same subjects (Condition *B*). Facial expression changes along the happy-angry axis were therefore the low-dimensional pattern that is enriched in condition *A*. **(B)** The first $\alpha$ identified by cPCA's algorithm, $\alpha = 0.2$, yields loadings similar to the first PCA dimensions, *i.e.* eigenfaces. **(C)** Projecting the faces onto the cPCs reveals clusters unrelated to the happy and angry facial expressions in condition *A*, indicating an incorrect solution. **(D)** cPCA with $\alpha = 1.7$ correctly reveals features associated with the expected facial expressions in condition *A*, such as furrowed eyebrows and the region

around the mouth and nose. **(E)** Projecting the faces onto these cPCs reveals the separation of happy and angry faces along the first cPC. Importantly, it was only possible to determine this answer was correct because we knew the labels in advance. **(F)** Dimensions identified by gcPCA correctly reflect features related to the facial expressions in condition *A*. **(G)** Projecting the faces onto the first two gcPCs also reveals the separation of happy and angry faces along the first gcPC.

useful for two reasons: 1) the categorical separation of facial expressions allows an easy evaluation of the contrastive methods, and 2) the dimensions can be visually inspected for features that are being discovered by the method.

We first applied cPCA to the two conditions and used its automatic $\alpha$ selection algorithm to pick two different $\alpha$ values. The automatic $\alpha$ selection algorithm developed by [14] finds representative $\alpha$'s that yield different cPC embeddings so that the investigator can choose the appropriate $\alpha$ value. The algorithm is explained fully in the original paper (see supplementary methods - algorithm 2 [14]). Briefly, it calculates cPCA for an array of different $\alpha$'s (default: 40 different $\alpha$ values ranging from 0.01 to 1000 spaced on a log scale), defining a subspace with the top $k$ cPCs (default: 2 dimensions), and calculating an affinity matrix between subspaces of different $\alpha$'s. This affinity matrix is then clustered to find $p$ clusters, and the medoid of each cluster is a candidate $\alpha$.

The first value returned is $\alpha = 0.2$, and we can see that the first two cPC loadings resemble "eigenfaces" [18], the largest principal components of facial images (Fig 2B). This suggests that this $\alpha$ is too small, leading cPCA to extract the highest-variance components in condition *A*. Looking at the individual faces projected on these cPCs, two clusters can be identified that separate cleanly along cPC1 (Fig 2C). However, this solution is incorrect because these clusters do not reflect the two facial expressions comprising condition *A*. Instead, they represent skin color, which should account for equal variance in both datasets because images of the same subjects are present in both conditions. For the second $\alpha$ value returned ($\alpha = 1.7$), the cPC loadings exhibit features specific to condition *A* (Fig 2D), and data projected onto these cPCs recovers the different expressions (Fig 2E). It is important to note that if we did not have the class labels *a priori*, we could easily believe $\alpha = 0.2$ was the correct answer because it produces better clustering than for $\alpha = 1.7$ (Fig 2C, 2E). Using gcPCA v3 (asymmetric, non-orthogonal) also reveals features specific to condition *A* (Fig 2F), and the two expressions in the dataset can be distinguished by their projection onto gcPC$_1$ (Fig 2G), without the requirement of fine-tuning a hyperparameter. gcPCA accomplishes this by extracting axes of increased variance in unsupervised fashion (S1C Fig), not by clustering or classifying the faces in a supervised manner as with LDA (S1B Fig).

## Applying gcPCA to neurophysiological recordings reveals hippocampal replay without *a priori* knowledge of replay content

A key application for gcPCA is neuronal recordings, which frequently contain activity from hundreds of individual neurons [19,20]. A well-studied neurophysiological phenomenon is hippocampal replay, in which hippocampal neurons encoding spatial trajectories traversed during a behavioral task "replay" the same activity patterns in post-task periods [21]. Note that spatial location is a continuous variable, so we are not expecting gcPCA or cPCA to find dimensions that cluster the activity into different groups, as with the facial expression data. Instead, we are hoping to see replay of the firing patterns that encode the linear track the animal recently explored [21,22]. We analyzed a previously published dataset recorded from hippocampal CA1 [23] where rats learn the location of an aversive air puff on a linear track. The

air puff is only delivered when the rat is running in one direction, called the danger run, and the other direction is the safe run (Fig 3A). In this dataset, [23] used traditional methods to demonstrate that hippocampal neurons exhibit reactivation of task-related activity patterns in post-task activity. These methods use activity patterns during task performance to define templates, then test whether activity patterns matching those templates are more common in post-task activity than pre-task activity [21,22,24].

We tested whether gcPCA could extract hippocampal replay in template-free fashion by contrasting post-task activity (condition *A*) with pre-task activity (condition *B*) (Fig 3B). We then compared the results of gcPCA to a template derived from activity during the task. To make this template, we performed traditional PCA on neuronal activity during running on the track. In these plots, the representations of the start, puff, and stop locations occupy distinct regions of PCA space, and the activity on the track forms bundles of trajectories connecting these points (Fig 3C). Since the PCA used data from all trials, danger and safe runs both exhibit this structure in PCA space. Strikingly, projecting the same runs onto the top gcPCs yielded qualitatively similar results (Fig 3D), suggesting that gcPCA (v4, see Methods) is extracting replay of task-related activity. Using PCA on either the pre- or post-task activity failed to extract this structure (Fig 3E), indicating it can only be found by contrasting POST with PRE. gcPCA was thus able to extract signatures of replay without prior knowledge of the task-related activity, which may prove useful in many analogous situations in which the experimenter does not have prior knowledge of the pattern they are searching for. Orthogonal and sparse gcPCA also yielded similar results (S3 Fig).

Using cPCA, it was not straightforward to detect replay. When we requested the cPCA algorithm evaluate all the dimensions ($k = 48$), the $\alpha$ values identified by the automatic selection did not reveal any obvious task-related spatial structure (Fig 3F – left, $\alpha$ values 0.4 and 942.7). When we requested a smaller set of dimensions ($k = 2$), the automatic $\alpha$ selection returned several different values, with one of them ($\alpha = 1.4$) revealing task-related structure (Fig 3F - right). This reveals that although the only explicit hyperparameter for cPCA is $\alpha$, the automatic $\alpha$ selection algorithm depends on $k$, the number of components requested, thus constituting a second *de facto* hyperparameter.

## Applying sparse gcPCA to scRNA-seq data prioritizes disease genes and reveals disease heterogeneity in type II diabetes

Another key application of gcPCA is high-dimensional omics datasets, which often compare two conditions (*e.g.* disease vs. healthy) to identify a list of features (*e.g.* genes) that differ between conditions. Sparse gcPCA is well-suited for this task because the sparsification narrows down the list of genes to the most important contributors. We therefore compared the performance of sparse gcPCA v4 to traditional sparse PCA [3,16] in published single-cell RNA sequencing data from pancreatic beta cells biopsied from patients with type II diabetes (T2D) or healthy controls [25]. We first performed sparse PCA separately on each condition and plotted the cells from both conditions in both PCA spaces (Fig 4A-4B). Cells from different subjects tended to aggregate together, and this pattern was largely preserved in either PCA space. This is because T2D and control cells exhibited similar covariance structure, with the loadings on the top PCs highly correlated (Fig 4D). In other words, sparse PCA extracts patterns that are shared between the conditions, finding little difference between them. Notably, LDA also failed to detect any differences between T2D and control conditions (Fig 4C), indicating that the average cellular transcriptional profile is the same in both conditions. Despite

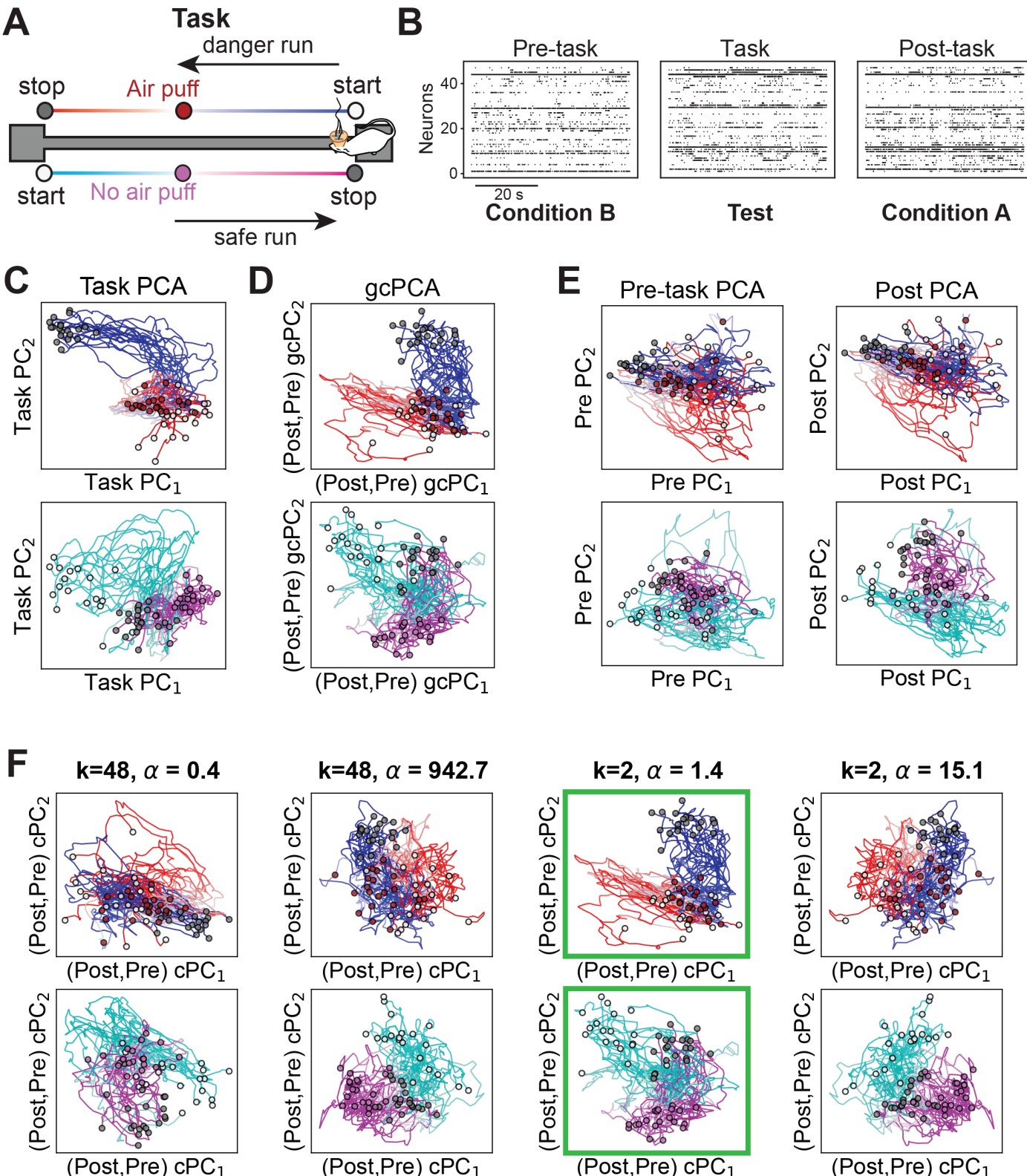

**Fig 3. gcPCA v4 correctly identifies hippocampal replay in neurophysiological data without prior knowledge of replay content. (A)** In [23], rats were trained to traverse a linear track where one direction has an aversive air puff and the other does not. Rats learned the location of the air puff and which direction was dangerous or safe. **(B)** Girardeau et al. (2017) recorded hippocampal neurons in pre- and post-task periods and used activity recorded during the task as a template to determine that that activity was "replayed" during post-task. We reanalyzed their published data using post-task activity as condition $A$ and pre-task activity as condition $B$ to identify the dimensions most enriched in post-task activity without taking task-related activity into account. **(C)** To visualize task-related activity,

we first performed PCA on task data, then projected activity during runs onto the first two PCs. This revealed mostly separate representations of the start/puff/stop locations, connected by bundles of trajectories. **(D)** We next applied gcPCA to post- and pre-task data, then projected activity during runs onto the first two gcPCs, which are enriched in post-task. This readily identifies a structure similar to that seen by the task PCA (panel C), suggesting that gcPCA is extracting replay during the post-task period. **(E)** PCA on the pre-task (left) or post-task (right) activity yields representations that are similar to each other but not activity during the task. This suggests that PCA is extracting shared components rather than replay. **(F)** cPCA requires hyperparameter searches yielding multiple solutions, with no indication of which is correct. The automatic $\alpha$ selection algorithm from cPCA returns various $\alpha$ values depending on the number of cPCs requested (parameter $k$). *Left Two Columns* Representative $\alpha$ values returned with $k$=48 cPCs. cPC$_{1-2}$ are the dimensions most enriched in post-task. No discernible spatial structure is identified, indicating that replay was not detected. *Right Two Columns* With $k$=2, the $\alpha$ values returned are of different magnitudes, and one of them ($\alpha$ = 1.43, green boxes) reveals spatial structure related to the task, suggestive of hippocampal replay.

the fact that the sparse PCs from each condition are correlated, sparse PCA of the T2D condition alone still extracts several known T2D-linked genes (Fig 4E, bold): *MPV17* [26], *SMCO4* [27], *DDIT3* [28,29], *NFE2L2* [30], *G6PC2##1* [31], and *SPPL2A* [32].

We next used sparse gcPCA v4 on the same datasets with T2D beta cells as condition *A* and control beta cells as condition *B*. gcPC$_1$ and gcPC$_2$ therefore represented axes along which T2D cells vary more, and gcPC$_{last}$ and gcPC$_{last-1}$ were axes along which control cells vary more. We found that T2D cells from different subjects formed two distinct clusters in the gcPC$_{1,2}$ subspace, possibly reflecting disease heterogeneity [33] (Fig 4F, left), but no clustering was evident for T2D cells in the gcPC$_{last, last-1}$ subspace (Fig 4F, right). Control cells did not exhibit clustering in either the gcPC$_{1,2}$ subspace (Fig 4G, left) or the gcPC$_{last, last-1}$ subspace (Fig 4G, right). This is because the gcPCs overrepresented in the T2D and control cells are largely uncorrelated (Fig 4H), indicating that gcPCA is successfully extracting patterns that differ between conditions. As expected, the top sparse gcPCs for T2D cells were weakly correlated with the top sparse PCs for T2D cells (Fig 4I), with only one of the top 20 T2D genes found with sparse gcPCA also found with sparse PCA (Fig 4J, *DDIT3*). A greater proportion of the genes in the sparse gcPC$_{1,2}$ gene list have been previously implicated in T2D (Fig 4J), including *TNFRSF12A* [34], *TFF3* [35], *IMMP2L* [36], *DDIT3* [28,29], *PPP1R1A* [37]), and *TMEM163* [38]. Notably, the *TMEM176A/B* genes were only recently discovered to be involved in T2D by [25], who demonstrated experimentally that these genes are functionally important for T2D-related beta-cell function.

## Materials and methods

### Generalized contrastive PCA

Our motivation for the following method stems from eliminating the necessity of the free parameter $\alpha$ in the contrastive PCA method. To accomplish this, we introduce a normalization factor to mitigate the bias toward high-variance dimensions. We will summarize the process of calculating the gcPCs using gcPCA v4 as an example, but v2 and v3 are analogous. gcPCA v4 has the following objective function, as shown in equation (eq. 8):

$$\underset{\mathbf{x}\,:\,\mathbf{x}^\mathsf{T}\mathbf{x}=1}{\arg\max} \frac{\mathbf{x}^\mathsf{T}(\mathbf{C}_A - \mathbf{C}_B)\mathbf{x}}{\mathbf{x}^\mathsf{T}(\mathbf{C}_A + \mathbf{C}_B)\mathbf{x}} \tag{8}$$

A potential problem of this objective function is the denominator creating numerical instability if the denominator covariance matrix has eigenvectors with eigenvalues equal to (or sufficiently close to) zero. This can occur either if the features are strongly correlated or if the data is rank-deficient, *e.g.* scRNA-seq data containing fewer cells than genes or two-photon imaging data containing fewer frames than pixels. To address this, we consider only the principal subspace of the covariances matrices, *i.e.* the subspace spanned by eigenvectors with

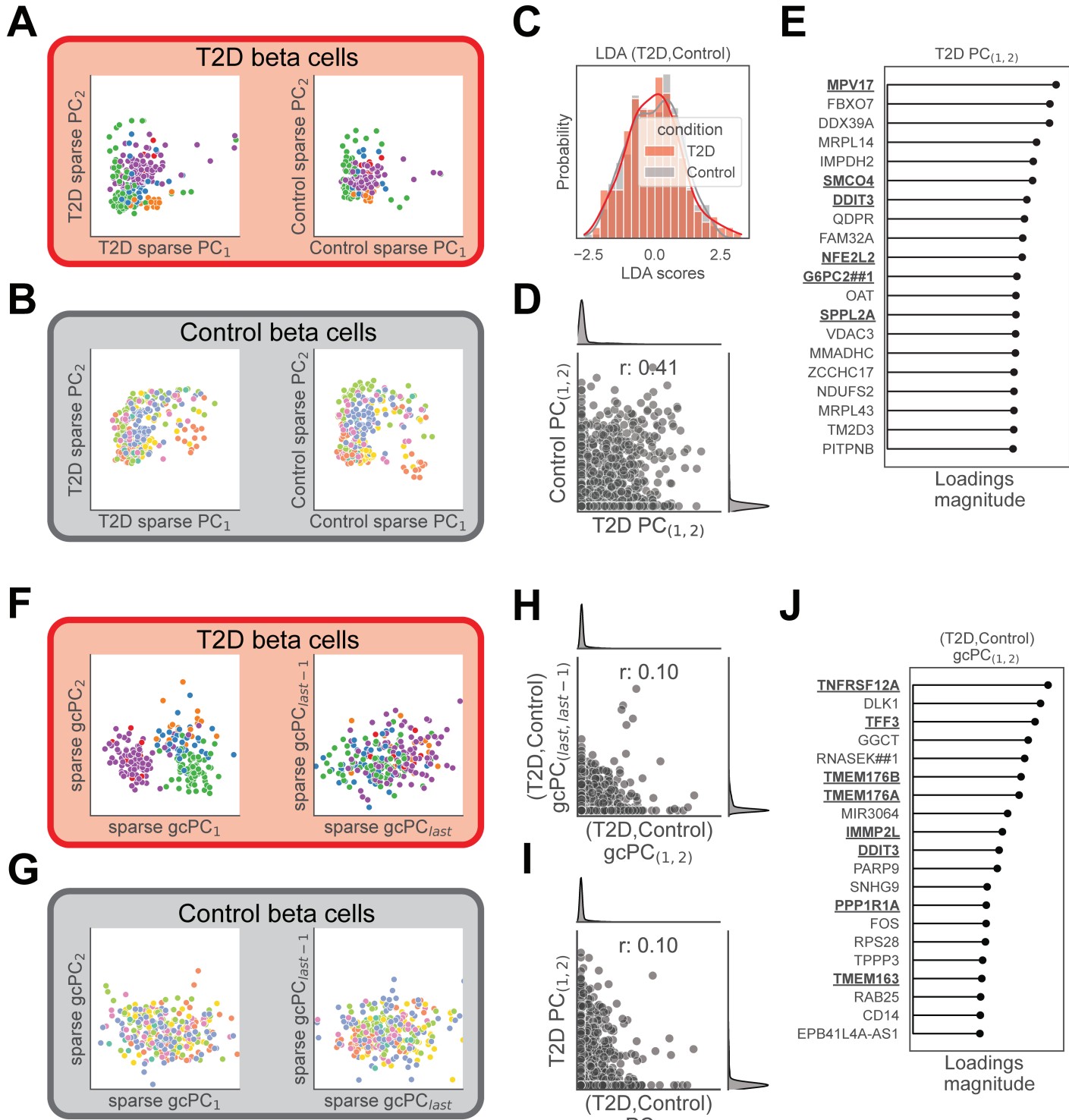

**Fig 4. Sparse gcPCA v4 reveals possible disease heterogeneity in type II diabetes.** To test sparse gcPCA, we compared it to traditional sparse PCA in the analysis of published scRNA-seq data from isolated pancreatic cells from type II diabetes (T2D) patients (red) and healthy controls (gray) [25]. **(A)** We first performed sparse PCA separately on the T2D and Control cells. In the T2D PCA space (left), T2D beta cells from the same subjects (indicated by color) tended to cluster together. This structure was largely retained in the Control PCA space (right), suggesting that sparse PCA extracts sources of variance that are common to both groups. **(B)** This same pattern was observed in Control cells, which also clustered together by subject identity in both PCA spaces. **(C)** Linear Discriminant Analysis (LDA) was unable to distinguish T2D cells from Control cells, indicating that the average gene expression profile does not differ between groups. **(D)** PCA loadings from Control or T2D cells were strongly correlated (r = 0.41), suggesting that sparse PCA on either group alone extracts sources of variability that are shared in both groups. **(E)** The top 20 genes identified by sparse PCA on T2D cells contain several known diabetes-related genes (bold). **(F)** We next performed sparse gcPCA 4 contrasting the T2D and Control cells. In the sparse $gcPC_{1,2}$ space (overrepresented in T2D cells, left), T2D cells from different subjects separated into distinct

subgroups, possibly revealing disease heterogeneity. This structure was not observed in the sparse $gcPC_{last,last-1}$ space (overrepresented in Control cells, right), indicating that gcPCA is extracting sources of variability that differ between groups. **(G)** Control cells in the sparse $gcPC_{last,last-1}$ space (right) did not exhibit the heterogeneity found in T2D cells, and clustering by subject identity was not preserved in the sparse $gcPC_{1,2}$ space (left). **(H)** The correlation between loadings for the top and bottom gcPCs was weak (r = 0.10), indicating that gcPCA is extracting sources of variance that differ between the T2D and Control cells. **(I)** Loadings for the top T2D PCs and gcPCs were less correlated (r = 0.10) than the T2D PC and Control PC loadings (r = 0.41, panel D), suggesting that PCA is extracting shared sources of variance, but gcPCA is extracting different sources. **(J)** The top 20 T2D genes extracted by gcPCA contained several genes previously implicated in T2D (bold), including *TMEM176A/B*, that were previously demonstrated as functionally important for T2D by Martínez-López et al. (2023).

non-zero eigenvalues. To project the matrices into this subspace, we first make a projection matrix $\mathbf{J}$ composed of the first $k$ principal components of the row-wise concatenated datasets $A$ and $B$, where $k$ is chosen so that the principal components that are effectively zero due to limited numerical precision are excluded from the analysis. We then substitute $\mathbf{x}$ using $\mathbf{x} = \mathbf{J}\gamma$, yielding:

$$\underset{\gamma\,:\,\gamma^\mathsf{T}\gamma=1}{\arg\max} \quad \frac{\gamma^\mathsf{T}\mathbf{J}^\mathsf{T}(\mathbf{C}_A - \mathbf{C}_B)\mathbf{J}\gamma}{\gamma^\mathsf{T}\mathbf{J}^\mathsf{T}(\mathbf{C}_A + \mathbf{C}_B)\mathbf{J}\gamma} \tag{9}$$

$\mathbf{J}$ guarantees that $\mathbf{J}^\mathsf{T}(\mathbf{C}_A + \mathbf{C}_B)\mathbf{J}$ in the denominator is positive definite, allowing us to find a symmetric matrix $\mathbf{M}$ that is its square root, yielding equation (eq. 10):

$$\underset{\gamma\,:\,\gamma^\mathsf{T}\gamma=1}{\arg\max} \quad \frac{\gamma^\mathsf{T}\mathbf{J}^\mathsf{T}(\mathbf{C}_A - \mathbf{C}_B)\mathbf{J}\gamma}{\gamma^\mathsf{T}\mathbf{M}^\mathsf{T}\mathbf{M}\gamma} \tag{10}$$

Let $\mathbf{y} = \mathbf{M}\gamma$, yielding:

$$\underset{\mathbf{y}\,:\,\mathbf{y}^\mathsf{T}\mathbf{y}=1}{\arg\max} \quad \frac{\mathbf{y}^\mathsf{T}\mathbf{M}^{-1}\mathbf{J}^\mathsf{T}(\mathbf{C}_A - \mathbf{C}_B)\mathbf{J}\mathbf{M}^{-1}\mathbf{y}}{\mathbf{y}^\mathsf{T}\mathbf{y}} \tag{11}$$

This optimization problem can be solved with the eigendecomposition of the numerator matrix to find $\mathbf{Y}$. The vectors in $\mathbf{X}$ are then calculated using equation (eq. 12):

$$\mathbf{X} = \mathbf{J}\mathbf{M}^{-1}\mathbf{Y} \tag{12}$$

The column vectors in $\mathbf{X}$ are referred to as gcPCs, following the term cPCs used in [14]. The other versions of gcPCA can be solved in the exact same way, by replacing $\mathbf{C}_A - \mathbf{C}_B$ with $\mathbf{C}_A$ (v2) or replacing $\mathbf{C}_A + \mathbf{C}_B$ with $\mathbf{C}_B$ (v2 and v3).

## Orthogonality constraint

The core idea of gcPCA, as described above, is to use $\mathbf{J}\mathbf{M}^{-1}$ to map the numerator $(\mathbf{C}_A - \mathbf{C}_B)$ from the original feature space into a weighted inner product space that is weighted by the inverse of the denominator. This warps the space so that the variance in $(\mathbf{C}_A + \mathbf{C}_B)$ is equal in all directions. This is directly analogous to the whitening transformation used to calculate Mahalanobis distances, but instead of calculating distances, we diagonalize $\mathbf{M}^{-1}\mathbf{J}^\mathsf{T}(\mathbf{C}_A - \mathbf{C}_B)\mathbf{J}\mathbf{M}^{-1}$ to find the gcPCs. Consequently, the gcPCs are orthogonal in this normalized feature space, not the original feature space. To calculate gcPCs that are orthogonal in the original feature space, we first take the largest eigenvector of $\mathbf{X}$ as the first gcPC. We then apply an orthogonality constraint by iteratively deflating matrix $\mathbf{J}$ to remove the subspace spanned by the gcPCs (*i.e.* columns of $\mathbf{X}$) that have already been found. At each step, we compute the eigenvector $\mathbf{x}$ of equation 11 with either the smallest or largest eigenvalue (alternating between them), then project it into the feature space with equation 12 and concatenate

it column-wise into the growing matrix **X**. To deflate **J**, we first regress out the **x** from **J**, as shown in equation 13:

$$\hat{\mathbf{J}} = \mathbf{J} - \mathbf{x}\mathbf{x}^\mathsf{T}\mathbf{J} \tag{13}$$

We then use SVD to get the left singular vectors of $\hat{\mathbf{J}}$, and we define $\tilde{\mathbf{J}}$ as the first $n-i$ of these (on the $i$-th iteration). $\tilde{\mathbf{J}}$ serves as an orthonormal basis for the subspace of **J** that is orthogonal to the columns of **X** that have already been calculated, and we use $\tilde{\mathbf{J}}$ as the new **J** in eq. 9 for the next iteration. This process continues until $k$ gcPCs are found, which can be the number of features in the dataset, the minimum rank of the conditions, or a number specified by the user.

## Sparse gcPCA

We developed an extension for sparse gcPCA by using a similar approach to sparse PCA [3] and sparse cPCA [15]. We will first review the sparse PCA framework, then explain how we adapt it for gcPCA.

**Sparse PCA method.** Sparse PCA was first proposed as a reinterpretation of PCA as a regression problem. In brief, given a matrix $\mathbf{X}_{n\times k}$ containing the first $k$ ordinary PCs, PCA can be seen as minimizing the following objective:

$$\underset{\mathbf{X}_{n\times k}:\mathbf{X}^\mathsf{T}\mathbf{X}=I}{\arg\min} ||\mathbf{D} - \mathbf{D}\mathbf{X}\mathbf{X}^\mathsf{T}||^2 \tag{14}$$

Where **D** is the data matrix of size features $\times$ samples. To achieve sparse loadings in the first $k$ PCs, Zou et al. (2006) [3] propose the use of elastic net regularization, as shown in the following objective function:

$$\underset{\mathbf{Y}_{n\times k},\mathbf{B}_{n\times k}:\mathbf{Y}^\mathsf{T}\mathbf{Y}=I}{\arg\min} ||\mathbf{D} - \mathbf{D}\mathbf{B}\mathbf{Y}^\mathsf{T}||^2 + \kappa \sum_{j=1}^{k} ||\beta_j||^2 + \sum_{j=1}^{k} \lambda ||\beta_j||_1 \tag{15}$$

where **B** is the matrix containing the sparse PCs $\beta_j$, and **Y** is a column-wise matrix that projects data from the sparse PC space to the feature space. The elastic net is the combination of the ridge penalty ($\kappa \sum_{j=1}^{k} ||\beta_j||^2$) and the lasso penalty ($\sum_{j=1}^{k} \lambda ||\beta_j||_1$). The ridge penalty is used to correct a rank-deficient matrix **D** for numerical purposes. In Zou et al. (2006) [3], the same $\kappa$ was used for all $k$ components while using a different $\lambda$ for every component. To numerically solve equation 15, Zou et al. (2006) [3] use an alternating algorithm in which **Y** is held constant as we solve for **B**, then **B** is held constant while we update **Y**, and this is repeated until the algorithm converges. **Y** is initially set to be equal to the ordinary PCs (**X**). Next, we find each column of **B** by the following elastic net regression:

$$\hat{\beta}_j = \underset{\beta_j}{\arg\min} ||\mathbf{D}\mathbf{y}_j - \mathbf{D}\beta_j||^2 + \kappa ||\beta_j||^2 + \lambda ||\beta_j||_1 \tag{16}$$

Next, **B** is fixed, meaning the penalty terms can be ignored, and the new **Y** is defined as:

$$\underset{\mathbf{Y}^\mathsf{T}\mathbf{Y}=I_{k\times k}}{\arg\min} ||\mathbf{D} - \mathbf{D}\mathbf{B}\mathbf{Y}^\mathsf{T}||^2 \tag{17}$$

The solution to 17 can be found by a reduced rank form of Procrustes rotation. Using SVD, we find

$$(\mathbf{D}^\mathsf{T}\mathbf{D})\mathbf{B} = \mathbf{USV}^\mathsf{T} \tag{18}$$

We then set $\hat{\mathbf{Y}} = \mathbf{UV}^\mathsf{T}$. To solve eq. 16, Zou et al. (2006) [3] has shown that is only necessary to know the Gram matrix $\mathbf{D}^\mathsf{T}\mathbf{D}$. For a fixed $\mathbf{Y}$, finding $\beta_j$ is equivalent to minimizing:

$$\begin{aligned}
&||\mathbf{D}\mathbf{y}_j - \mathbf{D}\beta_j|| + \kappa||\beta_j||^2 + \lambda||\beta_j||_1 \\
&= (\mathbf{y}_j - \beta_j)^\mathsf{T}\mathbf{D}^\mathsf{T}\mathbf{D}(\mathbf{y}_j - \beta_j) + \kappa||\beta_j||^2 + \lambda||\beta_j||_1
\end{aligned} \tag{19}$$

If the covariance matrix of $\mathbf{D}$ is known (denoted below as $\Sigma$), the term $\mathbf{D}^\mathsf{T}\mathbf{D}$ can be replaced with $\Sigma$. For solving the eq. 16, the $\mathbf{D}$ matrix can be replaced by $\Sigma^{\frac{1}{2}}$, which is the matrix square root of $\Sigma$. This results in the updated equation 20

$$\hat{\beta}_j = \arg\min_{\beta_j} ||\Sigma^{\frac{1}{2}}\mathbf{y}_j - \Sigma^{\frac{1}{2}}\beta_j||^2 + \kappa||\beta_j||^2 + \lambda||\beta_j||_1 \tag{20}$$

**Sparse gcPCA method.** We implemented sparse gcPCA by adapting the sparse PCA method presented and following similar steps proposed in [15]. For sparse gcPCA, the covariance matrix $\Sigma$ is replaced with the matrix $\Theta$, which reflects the appropriate objective function for the gcPCA version used. Following gcPCA objective function 11, $\Theta = \mathbf{M}^{-1}\mathbf{J}^\mathsf{T}(\mathbf{C}_A - \mathbf{C}_B)\mathbf{J}\mathbf{M}^{-1}$, where $\mathbf{J}$ is the projection basis for the principal subspace and $\mathbf{M}$ is the matrix square root of $\mathbf{J}^\mathsf{T}(\mathbf{C}_A + \mathbf{C}_B)\mathbf{J}$. For version 1 (equivalent to cPCA), we instead use $\Theta = \mathbf{C}_A - \alpha\mathbf{C}_B$, and for versions 2 and 3 we change $\Theta$ to match their respective objective functions, as mentioned previously. The components $\mathbf{X}$ are the gcPCs identified by the ordinary gcPCA algorithm. Following the numerical solution for sparse PCA presented before, the sparse gcPCA is obtained by the following alternating algorithm until convergence:

**B given Y:** Each sparse gcPC (denoted here as $\beta_j$) was found according to the following elastic net solution:

$$\hat{\beta}_j = \arg\min_{\beta_j} ||\Theta^{\frac{1}{2}}\mathbf{y}_j - \Theta^{\frac{1}{2}}\beta_j||^2 + \kappa||\mathbf{J}\mathbf{M}^{-1}\beta_j||^2 + \lambda||\mathbf{J}\mathbf{M}^{-1}\beta_j||_1 \tag{21}$$

where $\mathbf{y}_j$ is the $j^\text{th}$ gcPC and $\beta$ is the sparse gcPC. The ridge penalty $\kappa$ is used to fix rank-deficient matrices. To simplify our approach, we used the same $\lambda$ for all components. For gcPCA versions 2-4, $\beta$ is not in the original feature space, but in the feature space transformed by $\mathbf{J}\mathbf{M}^{-1}$. We therefore apply the regularization in the original feature space by using $\mathbf{J}\mathbf{M}^{-1}\beta$ instead of $\beta$ in the penalty terms, which ensures the loadings for the gcPCs are sparse. Eq. 21 can be solved through least angle regression, similar to [3], but our implementation uses variable projection [16], which is much more computationally efficient.

**Y given B:** Using fixed $\mathbf{B}$ in the normalized feature space, we can find a new $\hat{\mathbf{Y}}$ with a Procrustes rotation using SVD:

$$\Theta^{\frac{1}{2}}\Theta^{\frac{1}{2}}\mathbf{B} = \mathbf{USV}^\mathsf{T} \tag{22}$$

We can then determine $\hat{\mathbf{Y}} = \mathbf{UV}^\mathsf{T}$. These steps are repeated until loadings converge; fortunately, the problem is convex, and in practice we have not had problems with convergence. The sparse gcPCA loadings are defined as $\mathbf{J}\mathbf{M}^{-1}\mathbf{B}$. The main caveat with this approach is that

for gcPCA v3 or v4, the matrix $\Theta$ can have negative eigenvalues, which prevents the calculation of the square root matrix $\Theta^{\frac{1}{2}}$. Fortunately, these eigenvalues have a lower bound of -1, making the problem easy to address by using Tikhonov regularization to raise all of the eigenvalues to positive numbers, making $\Theta_+$ positive definite.

## Synthetic data generation

In the synthetic data, we generated two conditions with $1 \times 10^5$ samples and 100 dimensions. The dimensions were sampled from a Gaussian distribution (mu = 0 and sigma = 1) and then orthogonalized using singular value decomposition and picking the left singular vectors. In each condition, we created a pattern in the samples that was to be discovered. In condition $A$, we took dimensions 71 and 72 and drew the samples from a uniform distribution ($[0, 1]$). In dimension 71 we replaced any value from 0.3 and 0.7 with a different uniform distribution ($[0, 0.4]$). In dimension 72 all the values between 0.4 and 0.6 were replaced with another uniform distribution ($[0, 0.4]$). This created the square with a square hole in the middle in Fig 1A. The values were then offset by 0.5, the samples were sorted by the angle they formed in each dimension, calculated by the inverse tangent ($\tan^{-1} \frac{X_{71}}{X_{72}}$). The samples in each dimension were normalized by their $l_2$-norm. In condition $B$, we generated the samples of dimensions 81 and 82 from a uniform distribution ($[0, 1]$), sorted the sample values based on dimension 81, and rotated both dimensions by 45 degrees. This created the diamond shape seen in Fig 1A. The sample values were later normalized by their $l_2$-norm. The samples for all the other dimensions were drawn from a Gaussian distribution ($\mu = 0$ and $\sigma = 1$), and then normalized by their $l_2$-norm. The magnitude of each dimension was established as a line with a negative slope, starting at value 10 in the 1st dimension and ending at 0.001 in the 100th dimension. For condition $A$, we doubled the magnitude in dimensions 71 and 72, while in condition $B$ we doubled the magnitude in dimensions 81 and 82. Identifying these changes in magnitude is the goal of contrastive methods. Because the samples were drawn from a normal distribution, the dimensions will display correlations among them. To estimate the total variance explained by each dimension, we use a QR decomposition approach described in [3]. In brief, let $Z$ be a matrix containing scores of each dimension generated, the variance is usually calculated through $\text{tr}(Z^{\mathsf{T}}Z)$, where tr is the trace of the matrix. However, in correlated scores, this estimate is too optimistic. Using regression projection, it is possible to find the linear relationships of the dimensions and correct to find the adjusted total variance. Zou et al. (2006) [3] shows that this is equivalent to using QR decomposition in $Z$, such that $Z = QR$ where $Q$ is orthonormal and $R$ is upper triangular, and calculating the adjusted variance as follows:

$$\text{adjusted variance}_i = \sum_{j=1}^{k} R_{ij}^2 \tag{23}$$

## Face dataset

For the facial expression analysis, we used the Chicago Face Database [17]. This database consists of neutral and emotional expression faces, and for the analysis we used a subset of samples that had happy and angry faces alongside neutral ones. We used only male faces to reduce variability in feature positioning. The images used were cropped by an ellipse (length of 75 pixels and width of 45 pixels) centered in the face to focus on the facial expression rather than other features such as hairstyle or shoulders. Each sample image was flattened from a two-dimensional matrix to a vector, and all the flattened samples were then concatenated, resulting in a matrix of samples x features. Each feature was z-scored and normalized by its

$l^2$-norm. Condition *A* consisted of all the samples of happy and angry facial expressions, while condition *B* samples were neutral expressions.

## Hippocampal electrophysiology data

We used a previously published hippocampal electrophysiology dataset, with the experimental details listed in the original publication [25,39]. In brief, Long-Evans rats were implanted with silicon probes in the dorsal hippocampus CA1 region (either left or right hemisphere), and neuronal activity was isolated through automatic spike sorting and manually curated. Animals were trained to collect water rewards at the end of a linear track, and an air puff was introduced at a fixed location for every lap, in only one of the directions. Recordings consisted of task, where the animal learned the air puff location, and periods of pre- and post-task activity. For testing the contrastive methods, we used pre-task recordings as condition *B* and post-task recordings as condition *A*. For our analysis, we only used neurons that had a minimum firing rate of 0.01 spikes/s during the task. We binned the neural data using a bin size of 10 ms and smoothed using a rolling average with a Gaussian window of size 5 bins. The data was then z-scored and normalized by the norm before applying the contrastive methods. The task data was then projected on the contrastive dimensions for evaluation. The same normalized data was used for the PCA analysis.

## Pancreatic single-cell RNA sequencing

For the single-cell RNA sequencing data analysis, we used a previously published dataset [25], available at GEO accession GSE153855, consisting of scRNA-seq data from human pancreatic islet cells from patients with type II diabetes and healthy controls. We used the annotated dataset to identify the beta cells, which were identified previously by the authors [25]. For condition *A* we used the beta cells from subjects that had type II diabetes, and for condition *B* we used the beta cells from healthy patients. We used the expression values in reads per kilobase of the gene model and million mappable reads (RPKMs). The values were log-transformed, and all the features were centered before the analysis. We used the same set of genes used by Martínez-López et al. (2023) [25]. To determine the top genes found by each method (sparse PCA or sparse gcPCA), we multiply the loadings on each dimension by their eigenvalue and calculate the $l^2$-norm, returning a loading magnitude for each gene.

## Discussion

Discovering low-dimensional patterns that differ between conditions in high-dimensional datasets is a crucial analysis in many research contexts. Here we present gcPCA, a method that achieves this by examining the covariance structure of the datasets to find dimensions that exhibit the largest relative changes in variance between conditions. This work builds on the pioneering insights in the development of cPCA [14] but solves cPCA's key problem, the requirement for the hyperparameter $\alpha$. Here we showed that the function of $\alpha$ is to compensate for bias toward high-variance dimensions in noisy, finitely-sampled data. Further, we showed how this can be circumvented by introducing a normalization factor to penalize high-variance dimensions. Previous work [14,15] has focused on developing and improving methods to find appropriate choices for $\alpha$, but with gcPCA we chose instead to eliminate the $\alpha$ hyperparameter entirely. The key advantage of this approach is not merely that it is computationally cheaper than scanning a range of $\alpha$'s; it is that in most real-world cases there is no way to know whether a given choice of $\alpha$ yields a correct solution.

We wish to address a common point of confusion by reiterating how gcPCA differs from LDA or Partial Least Squares (PLS) regression. LDA and PLS find patterns optimally distinguishing two datasets, but gcPCA finds patterns that exhibit more within-dataset variability in one dataset than another (S1A Fig). As a fictitious example, LDA might find a height dimension distinguishing a university basketball team from the general public because the average basketball player is taller than the average person. In contrast, gcPCA would likely find an age/education-level dimension because those features exhibit more variability across members of the general public than across members of a university team. Despite the simplicity of this approach, it reveals interesting phenomena in high-dimensional biological data such as hippocampal replay (Fig 3) or transcriptomic heterogeneity in disease states (Fig 4).

gcPCA has two main caveats: first, unlike ordinary PCs or cPCs, gcPCs are not orthogonal in the original feature space, but rather in the normalized feature space. This is because gcPCA applies the normalization to the data before finding the gcPCs, which is optimal for contrasting $A$ and $B$. However, our toolbox includes versions of gcPCA that are constrained to be orthogonal in feature space (v2.1, v3.1, and v4.1), which comes at increased computational cost because a new eigendecomposition must be performed for each gcPC. In most cases using the constraint makes little difference to the end results (S3A-S3D Fig), but the difference will be greater in datasets where some dimensions exhibit much more variance than others, increasing the amount of normalization required. The second caveat is that the normalization factor introduces the possibility of numerical instability if the denominator matrix is rank-deficient, meaning it has dimensions with zero (or near-zero) variance that create division by zero. However, the implementation of gcPCA in the toolbox automatically excludes these dimensions if they exist.

One limitation of this study is that we did not establish rigorous criteria for what constitutes "noisy and finitely-sampled" data likely to benefit from penalizing high-variance dimensions (Fig 1), but almost all real-world biological datasets likely qualify to some degree. There may also be situations in which cPCA's $\alpha$ could be a feature, rather than a bug, if the investigator has prior knowledge that the patterns of interest will lie in high- or low-variance dimensions. Choosing an appropriate $\alpha$ could then intentionally bias the analysis in favor of the results of interest. In such cases, it would be relatively straightforward to extend gcPCA by adding a parameter that accomplishes a similar result by adjusting the eigenspectrum of the denominator matrix in the objective function, *e.g.* $(\mathbf{C}_A + \mathbf{C}_B)$. There are also other extensions that could be added, such as contrasting more than two conditions or incorporating nonlinearity, which can be relevant to specific data problems. However, we leave the development of such tools for future efforts.

The biological sciences are currently undergoing an explosion of technologies that produce high-dimensional datasets, including novel forms of microscopy and neuroimaging, high-speed video tracking, Neuropixels recordings, -omics approaches with single-cell resolution, and many others. In addition to the analyses of electrophysiological recordings or single-cell RNA sequencing data demonstrated here, gcPCA could be applied to any of these experimental modalities. We thus anticipate that the open-source gcPCA toolbox will provide a valuable resource facilitating a broad range of biological investigations that require contrasting two experimental conditions.

## Supporting information

**S1 Fig. gcPCA and Linear Discriminant Analysis (LDA) are complementary tools that solve distinct problems. (A)** LDA finds the dimension that optimally distinguishes conditions A and B by simultaneously maximizing the distance between them and minimizing the

inter-condition variance. PCA finds the dimensions that maximize the variance of condition A alone, with no regard to condition B. gcPCA finds dimensions that maximize the variance in one condition and minimize it in the other. The first gcPCs have maximal variance in A and minimal variance in B, and the last gcPCs have maximal variance in B and minimal variance in A. **(B)** When applied to the facial expression data from Fig 2, LDA is unable to distinguish condition A from B (vertical axis). However, LDA can distinguish happy from angry faces (horizontal axis) provided the data are labeled in advance. **(C)** gcPCA distinguishes happy and angry faces in the datasets from Fig 2 in unsupervised fashion. Note that the variance among neutral faces (gray) is minimized.
(TIF)

**S2 Fig. The range of cPCA $\alpha$ values yielding correct solutions can be very narrow. (A)** We generated synthetic data with enriched variance in both high and low variance dimensions. In condition A we enriched the variance in dimensions 19 and 92, and in condition B we enriched the variance in dimensions 31 and 82. This panel shows the finite and infinite data results for $\mathbf{C}_A - \mathbf{C}_B$. Stars represents the finite data value in the enriched dimensions. Even though the high variance dimensions are easy to detect with this method, the low variance ones are still occluded by spurious variability in high variance dimensions. **(B)** cPCA can reveal enriched high variance dimensions, but enriched low variance dimensions are hard to identify. **(C)** gcPCA can find all enriched dimensions simultaneously for conditions *A* and *B*. **(D)** The range of $\alpha$ values yielding the correct solution becomes narrow because the enriched dimensions have different absolute variance. **(E)** gcPCA correctly identifies all enriched dimensions in both conditions.
(TIF)

**S3 Fig. The gcPCA toolbox contains versions of gcPCA with constraints for orthogonality and sparsity. (A)** The gcPC covariance matrix for the hippocampal data in Fig 3 using gcPCA v4. The non-zero values off the diagonal reveal that, by default, gcPCs are not orthogonal in original feature space, though in practice they are usually quite close. **(B)** gcPC loadings for gcPCA v4. **(C-D)** The projection of task data onto the gcPCs exhibits spatial structure similar to PCs from the task data (Fig 3C). **(E)** Orthogonal gcPCA v4.1 finds gcPCs that are orthogonal in the original feature space, as shown by the absence of non-zero values off the diagonal. Note that the processing time is substantially longer. **(F)** For these datasets, the loadings for orthogonal gcPCA v4.1 are almost indistinguishable from standard gcPCA v4 (panel B). **(G-H)** Projecting task data onto the orthogonal gcPCs yields almost identical results to standard gcPCA v4 (panels C-D). **(I)** With sparse gcPCA, the sparsification of the gcPCs creates deviations from orthogonality. The processing time is also substantially longer. **(J)** Note that the loadings are sparser than in panels B and F, meaning that the most important features have been highlighted. **(K-L)** Task data projected onto the sparse gcPCs is somewhat distorted relative to panels C-D and G-H.
(TIF)

## Acknowledgments

We would like to thank Soyoun Kim, Ruben Coen-Cagli, Ehsan Sabri, Wenzhu Mowrey, Cleiton Lopes-Aguiar, Roman Huszár, and members of the Sjulson and Batista-Brito lab for valuable conversations and insightful comments on the manuscript.

## Author contributions

**Conceptualization:** Eliezyer Fermino de Oliveira, Lucas Sjulson.

**Data curation:** Eliezyer Fermino de Oliveira, Pranjal Garg.

**Formal analysis:** Eliezyer Fermino de Oliveira, Pranjal Garg, Lucas Sjulson.

**Funding acquisition:** Lucas Sjulson.

**Investigation:** Eliezyer Fermino de Oliveira, Pranjal Garg, Lucas Sjulson.

**Methodology:** Eliezyer Fermino de Oliveira, Lucas Sjulson.

**Project administration:** Lucas Sjulson.

**Resources:** Lucas Sjulson.

**Software:** Eliezyer Fermino de Oliveira, Pranjal Garg, Lucas Sjulson.

**Supervision:** Jens Hjerling-Leffler, Renata Batista-Brito, Lucas Sjulson.

**Visualization:** Eliezyer Fermino de Oliveira, Pranjal Garg.

**Writing – original draft:** Eliezyer Fermino de Oliveira, Lucas Sjulson.

**Writing – review & editing:** Eliezyer Fermino de Oliveira, Pranjal Garg, Jens Hjerling-Leffler, Renata Batista-Brito, Lucas Sjulson.

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
