## [Decision Letter · Decision Letter 0]

4 Nov 2024

PCOMPBIOL-D-24-01365Identifying patterns differing between high-dimensional datasets with generalized contrastive PCAPLOS Computational Biology Dear Dr. Sjulson, Thank you for submitting your manuscript to PLOS Computational Biology. After careful consideration, we feel that it has merit but does not fully meet PLOS Computational Biology's publication criteria as it currently stands. Therefore, we invite you to submit a revised version of the manuscript that addresses the points raised during the review process. Please submit your revised manuscript within 60 days Jan 04 2025 11:59PM. If you will need more time than this to complete your revisions, please reply to this message or contact the journal office at ploscompbiol@plos.org. Please include the following items when submitting your revised manuscript: * A rebuttal letter that responds to each point raised by the editor and reviewer(s). You should upload this letter as a separate file labeled 'Response to Reviewers'. This file does not need to include responses to formatting updates and technical items listed in the 'Journal Requirements' section below.* A marked-up copy of your manuscript that highlights changes made to the original version. You should upload this as a separate file labeled 'Revised Manuscript with Track Changes'.* An unmarked version of your revised paper without tracked changes. You should upload this as a separate file labeled 'Manuscript'. If you would like to make changes to your financial disclosure, competing interests statement, or data availability statement, please make these updates within the submission form at the time of resubmission. Guidelines for resubmitting your figure files are available below the reviewer comments at the end of this letter. We look forward to receiving your revised manuscript. Kind regards, Stefano PanzeriAcademic EditorPLOS Computational Biology Lyle GrahamSection EditorPLOS Computational Biology Feilim Mac GabhannEditor-in-ChiefPLOS Computational Biology Jason PapinEditor-in-ChiefPLOS Computational Biology  **Journal Requirements:** **Additional Editor Comments (if provided):****Reviewers' comments:** Reviewer's Responses to Questions

**Comments to the Authors:**

Reviewer #1: Uploaded as an attached pdf

Reviewer #2: In this paper, the authors propose a variant of contrastive PCA, an recently-developed algorithm that identifies differing patterns between two contrasted datasets. The authors' contribution is a set of algorithms that are parameter-free unlike the original algorithm and also more intuition into the outputs of the original algorithm its limitations and suggestions on how to overcome these. Application of these algorithms to three datasets showcases their features well. I found the paper informative, insightful, and well-written and have a few suggestions that could strengthen its impact and usability:

a) I found the distinction that the authors provide between discriminant analyses and cPCA very useful and intuitive for prospective users of the methodology. I wonder if the authors could also apply LDA-like analyses to one of the datasets they used here and compare/contrast the results with gcPCA to offer more intuition into these differences. This would be very informative for the readers to appreciate the usefulness of cPCA and gcPCA in particular here and understand when these are more suitable for their analyses over LDA.

b) Can we get some more intuition into what constitutes "infinite data" in this context? Is there a threshold that the authors have empirically determined? Or are all actual datasets finite?

c) Some more intuition into when orthogonality or sparsity would be useful in this context would help too. For example, what outputs would the orthogonal or sparse gcPCAs give for one of the examined datasets? How would these differ from the unconstrained ones? Are there other caveats in including such constraints, e.g. algorithm convergence or complexity?

**Have the authors made all data and (if applicable) computational code underlying the findings in their manuscript fully available?**

Reviewer #1: None

Reviewer #2: Yes

PLOS authors have the option to publish the peer review history of their article (what does this mean?). If published, this will include your full peer review and any attached files.

Reviewer #1: No

Reviewer #2: **Yes: **Ioannis Delis

 **Figure resubmission:**While revising your submission, please upload your figure files to the Preflight Analysis and Conversion Engine (PACE) digital diagnostic tool, https://pacev2.apexcovantage.com/. PACE helps ensure that figures meet PLOS requirements. To use PACE, you must first register as a user. Registration is free. Then, login and navigate to the UPLOAD tab, where you will find detailed instructions on how to use the tool. If you encounter any issues or have any questions when using PACE, please email PLOS at figures@plos.org. Please note that Supporting Information files do not need this step. If there are other versions of figure files still present in your submission file inventory at resubmission, please replace them with the PACE-processed versions. 
---

## [Decision Letter · Decision Letter 1]

23 Dec 2024

Dear Dr. Sjulson,

We are pleased to inform you that your manuscript 'Identifying patterns differing between high-dimensional datasets with generalized contrastive PCA' has been provisionally accepted for publication in PLOS Computational Biology.

Best regards,

Stefano Panzeri

Academic Editor

PLOS Computational Biology

Lyle Graham

Section Editor

PLOS Computational Biology

Reviewer's Responses to Questions

**Comments to the Authors:**

Reviewer #2: The new figures in the supplement are useful additions to the manuscript

**Have the authors made all data and (if applicable) computational code underlying the findings in their manuscript fully available?**

Reviewer #2: Yes

PLOS authors have the option to publish the peer review history of their article (what does this mean?). If published, this will include your full peer review and any attached files.

Reviewer #2: **Yes: **Ioannis Delis

---

## [Editor Report · Acceptance letter]

PCOMPBIOL-D-24-01365R1

Identifying patterns differing between high-dimensional datasets with generalized contrastive PCA

Dear Dr Sjulson,

I am pleased to inform you that your manuscript has been formally accepted for publication in PLOS Computational Biology. Your manuscript is now with our production department and you will be notified of the publication date in due course.

With kind regards,

Anita Estes
